# Thermostat-assisted continuously-tempered Hamiltonian Monte Carlo for Bayesian learning

**Rui Luo[1], Jianhong Wang[*1], Yaodong Yang[*1], Zhanxing Zhu[2], and Jun Wang[†1]**
[1]University College London, [2]Peking University

## Abstract

We propose a new sampling method, the thermostat-assisted continuously-tempered Hamiltonian Monte Carlo, for Bayesian learning on large datasets and multimodal distributions. It simulates the Nosé-Hoover dynamics of a continuously-tempered Hamiltonian system built on the distribution of interest. A significant advantage of this method is that it is not only able to efficiently draw representative i.i.d. samples when the distribution contains multiple isolated modes, but capable of adaptively neutralising the noise arising from mini-batches and maintaining accurate sampling. While the properties of this method have been studied using synthetic distributions, experiments on three real datasets also demonstrated the gain of performance over several strong baselines with various types of neural networks plunged in.

## 1 Introduction

Bayesian learning via Markov chain Monte Carlo (MCMC) methods is appealing for its inborn nature of characterising the uncertainty within the learnable parameters. However, when the distributions of interest contain multiple modes, rapid exploration on the corresponding multimodal landscapes w.r.t. the parameters becomes difficult using classic methods [7, 16]. In particular, given a large number of modes, some "distant" ones might be beyond the reach from others; this would potentially lead to the so-called *pseudo-convergence* [1], where the guarantee of ergodicity for MCMC methods breaks.

To make things worse, Bayesian learning on large datasets is typically conducted in an online setting: at each of the iterations, only a subset, i.e. a mini-batch, of the dataset is utilised to update the model parameters [24]. Although the computational complexity is substantially reduced, those mini-batches inevitably introduce noise into the system and therefore increase the uncertainty within the parameters, making it harder to properly sample multimodal distributions.

In this paper, we propose a new sampling method, referred to as the thermostat-assisted continuously-tempered Hamiltonian Monte Carlo, to address the aforementioned problems and to facilitate Bayesian learning on large datasets and multimodal posterior distributions. We extend the classic Hamiltonian Monte Carlo (HMC) with the scheme of continuous tempering stemming from the recent advances in physics [8] and chemistry [15]. The extended dynamics governs the variation on effective temperature for the distribution of interest in a continuous and systematic fashion such that the sampling trajectory can readily overcome high energy barriers and rapidly explore the entire parameter space. In addition to tempering, we also introduce a set of Nosé-Hoover thermostats [18, 11] to handle the noise arising from the use of mini-batches. The thermostats are integrated into the tempered dynamics so that the mini-batch noise can be effectively recognised and automatically neutralised. In short, the proposed method leverages continuous tempering to enhance the sampling efficiency, especially for multimodal distributions; it makes use of Nosé-Hoover thermostats to adaptively dissipate the instabilities caused by mini-batches so that the desired distributions can be recovered. Various experiments are conducted to demonstrate the effectiveness of the new method: it consistently outperforms several samplers and optimisers on the accuracy of image classification with different types of neural network.

---

[*]Equal
[†]Correspondence to: `j.wang@cs.ucl.ac.uk`

## 2 Preliminaries

We review HMC [6] and continuous tempering [8, 15], the two bases of our model, where the former serves as a *de facto* standard for Bayesian sampling and the latter is a state-of-the-art solution to the acceleration of molecular dynamics simulations on complex physical systems.

### 2.1 Hamiltonian Monte Carlo for posterior sampling

Bayesian posterior sampling aims at efficiently generating i.i.d. samples from the posterior $\rho(\boldsymbol{\theta}|\mathcal{D})$ of the variable of interest $\boldsymbol{\theta}$ given some dataset $\mathcal{D}$. Provided the prior $\rho(\boldsymbol{\theta})$ and the likelihood $\mathscr{L}(\boldsymbol{\theta};\mathcal{D})$ along with the dataset $\mathcal{D} = \{\boldsymbol{x}_i\}$ with $|\mathcal{D}|$ independent data points $\boldsymbol{x}_i$, the target posterior to generate samples from can be formulated as

$$\rho(\boldsymbol{\theta}|\mathcal{D}) \propto \rho(\boldsymbol{\theta})\mathscr{L}(\boldsymbol{\theta};\mathcal{D}) = \rho(\boldsymbol{\theta})\prod_{i}^{|\mathcal{D}|} \ell(\boldsymbol{\theta};\boldsymbol{x}_i), \text{ with the likelihood per data point } \ell(\boldsymbol{\theta};\boldsymbol{x}_i). \tag{1}$$

In a typical HMC setting [16], a physical system is constructed and connected with the target posterior in Eq. (1) via the system's potential, which is defined as

$$U(\boldsymbol{\theta}) = -\log \rho(\boldsymbol{\theta}|\mathcal{D}) = -\log \rho(\boldsymbol{\theta}) - \sum_{i=1}^{|\mathcal{D}|} \log \ell(\boldsymbol{\theta};\boldsymbol{x}_i) - \text{const}. \tag{2}$$

In this system, the variable of interest $\boldsymbol{\theta} \in \mathbb{R}^D$, referred to as the system configuration, is interpreted as the joint position of all physical objects within that system. An auxiliary variable $\boldsymbol{p}_\theta \in \mathbb{R}^D$ is then introduced as the conjugate momentum w.r.t. $\boldsymbol{\theta}$ to describe its rate of change. The tuple $\boldsymbol{\Gamma} = (\boldsymbol{\theta},\boldsymbol{p}_\theta)$ represents the state of the physical system that uniquely determines the characteristics of that system. A predefined constant matrix $\boldsymbol{M}_\theta = \text{diag}[m_{\theta_i}]$ specifies the masses of the objects associated with $\boldsymbol{\theta}$ and can be leveraged for preconditioning.

The energy function $H(\boldsymbol{\Gamma})$ of the physical system, referred to as the Hamiltonian, is essentially the sum of the potential in Eq. (2) and the conventional quadratic kinetic energy: $H(\boldsymbol{\Gamma}) = U(\boldsymbol{\theta}) + \boldsymbol{p}_\theta^\top \boldsymbol{M}_\theta^{-1}\boldsymbol{p}_\theta/2$. The Hamiltonian dynamics, i.e. the Hamilton's equations of motion, can be derived by applying the Hamiltonian formalism $[\dot{\boldsymbol{\theta}} = \partial_{\boldsymbol{p}_\theta}H, \dot{\boldsymbol{p}}_\theta = -\partial_{\boldsymbol{\theta}}H]$ to $H(\boldsymbol{\Gamma})$, where $\dot{\boldsymbol{\theta}}$ and $\dot{\boldsymbol{p}}_\theta$ denote the time derivatives.

The Hamiltonian dynamics, on one hand, describes the time evolution of system from a microscopic perspective. The principle of statistical physics, on the other hand, states in a macroscopic sense that given a physical system in thermal equilibrium with a heat bath at a fixed temperature $T$, the states $\boldsymbol{\Gamma}$ of that system are distributed as a particular distribution related to the system's Hamiltonian $H(\boldsymbol{\Gamma})$:

$$\pi(\boldsymbol{\Gamma}) = \frac{1}{Z_\Gamma(T)}e^{-H(\boldsymbol{\Gamma})/T}, \text{ with the normalising constant } Z_\Gamma(T) = \sum_{\boldsymbol{\Gamma}} e^{-H(\boldsymbol{\Gamma})/T}. \tag{3}$$

Such distribution is referred to as the canonical distribution. Note that by setting $T = 1$ and $U(\boldsymbol{\theta})$ as in Eq. (2), the canonical distribution in Eq. (3) can be marginalised as the posterior in Eq. (1).

### 2.2 Continuous tempering

In physical chemistry, continuous tempering [8, 15] is currently a state-of-the-art method to accelerate molecular dynamics simulations by means of continuously and systematically varying the temperature of a physical system. It extends the original system by coupling with additional degrees of freedom, namely the tempering variable $\xi \in \mathbb{R}$ with mass $m_\xi$ as well as its conjugate momentum $p_\xi \in \mathbb{R}$, which control the effective temperature of the original system in a continuous fashion via the Hamiltonian dynamics of the extended system. With a suitable choice of coupling function $\lambda(\xi)$ and a compatible confining potential $W(\xi)$, the Hamiltonian of the extended system can be designed as

$$H(\boldsymbol{\Gamma}) = \lambda(\xi)U(\boldsymbol{\theta}) + W(\xi) + \boldsymbol{p}_\theta^\top \boldsymbol{M}_\theta^{-1}\boldsymbol{p}_\theta/2 + p_\xi^2/2m_\xi, \tag{4}$$

where $\boldsymbol{\Gamma} = (\boldsymbol{\theta},\xi,\boldsymbol{p}_\theta,p_\xi)$ represents the state of the extended system with the position of the tempering variable $\xi$ and its momentum $p_\xi$ appended to the state of the original system $(\boldsymbol{\theta},\boldsymbol{p}_\theta)$. $\lambda(\xi) \in \mathbb{R}^+$ maps the tempering variable to a multiplier of temperature so that the effective temperature of the original system $T/\lambda(\xi)$ can vary; its domain $\text{dom}\lambda(\xi) \subset \mathbb{R}$ is a finite interval regulated by $W(\xi)$.

# 3 Thermostat-assisted continuously-tempered Hamiltonian Monte Carlo

We propose a sampling method, called the thermostat-assisted continuously-tempered Hamiltonian Monte Carlo (TACT-HMC), for multimodal posterior sampling in the presence of unknown noise. TACT-HMC leverages the extended Hamiltonian in Eq. (4) to raise and vary the effective temperature continuously; it efficiently lowers the energy barriers between modes and hence accelerates sampling. Our method also incorporates the Nosé-Hoover thermostats to effectively recognise and automatically neutralise the noise arising from the use of mini-batches.

## 3.1 System dynamics with the Nosé-Hoover augmentation

In solving for the system dynamics, we apply the Hamiltonian formalism to the extended Hamiltonian in Eq. (4), which requires the potential $U(\boldsymbol{\theta})$ and gradient $\nabla_{\boldsymbol{\theta}} U(\boldsymbol{\theta})$. We define hereafter the negative gradient of the potential $U(\boldsymbol{\theta})$ as the induced force $\boldsymbol{f}(\boldsymbol{\theta}) = -\nabla_{\boldsymbol{\theta}} U(\boldsymbol{\theta})$. Because the calculation of either $U(\boldsymbol{\theta})$ or $\boldsymbol{f}(\boldsymbol{\theta})$ involves the full dataset $\mathscr{D} = \{\boldsymbol{x}_i\}$, it is computationally expensive or even unaffordable to calculate the actual values for large $|\mathscr{D}|$. Instead, we consider the mini-batch approximations:

$$\tilde{U}(\boldsymbol{\theta}) = -\log\rho(\boldsymbol{\theta}) - \frac{|\mathscr{D}|}{|\mathcal{S}|} \sum_{k=1}^{|\mathcal{S}|} \log\ell(\boldsymbol{\theta};\boldsymbol{x}_{i_k}) \quad\text{and}\quad \tilde{\boldsymbol{f}}(\boldsymbol{\theta}) = \nabla_{\boldsymbol{\theta}}\log\rho(\boldsymbol{\theta}) + \frac{|\mathscr{D}|}{|\mathcal{S}|} \sum_{k=1}^{|\mathcal{S}|} \nabla_{\boldsymbol{\theta}}\log\ell(\boldsymbol{\theta};\boldsymbol{x}_{i_k}),$$

where $\boldsymbol{x}_{i_k}$ denotes the data point sampled from mini-batches $\mathcal{S} = \{\boldsymbol{x}_{i_k}\} \subset \mathscr{D}$ with the size $|\mathcal{S}| \ll |\mathscr{D}|$. It is clear that $\tilde{U}(\boldsymbol{\theta})$ and $\tilde{\boldsymbol{f}}(\boldsymbol{\theta})$ are unbiased estimators of $U(\boldsymbol{\theta})$ and $\boldsymbol{f}(\boldsymbol{\theta})$.

As we assume $\boldsymbol{x}_{i_k}$ to be mutually independent, $\tilde{U}(\boldsymbol{\theta})$ and $\tilde{\boldsymbol{f}}(\boldsymbol{\theta})$ are sums of $|\mathcal{S}|$ i.i.d. random variables, where the Central Limit Theorem (CLT) applies; the mini-batch approximations converge to Gaussian variables, i.e. $\tilde{U}(\boldsymbol{\theta}) \to \mathcal{N}(U(\boldsymbol{\theta}), v_U(\boldsymbol{\theta}))$ and $\tilde{\boldsymbol{f}}(\boldsymbol{\theta}) \to \mathcal{N}(\boldsymbol{f}(\boldsymbol{\theta}), \boldsymbol{V}_f(\boldsymbol{\theta}))$ with variances $v_U(\boldsymbol{\theta})$ and $\boldsymbol{V}_f(\boldsymbol{\theta})$. As random variables, $\tilde{U}(\boldsymbol{\theta})$ and $\tilde{\boldsymbol{f}}(\boldsymbol{\theta})$ inevitably inject noise into the system dynamics. We incorporate a set of independent Nosé-Hoover thermostats [18, 11] – apparatuses originally devised for temperature stabilisation in molecular dynamics simulations – to adaptively cancel the effect of noise. The system dynamics with the augmentation of thermostats – we call *Nosé-Hoover dynamics* – is formulated as

$$\frac{\mathrm{d}\boldsymbol{\theta}}{\mathrm{d}t} = \boldsymbol{M}_{\boldsymbol{\theta}}^{-1}\boldsymbol{p}_{\boldsymbol{\theta}}, \qquad \frac{\mathrm{d}\boldsymbol{p}_{\boldsymbol{\theta}}}{\mathrm{d}t} = \lambda(\xi)\tilde{\boldsymbol{f}}(\boldsymbol{\theta}) - \lambda^2(\xi)\boldsymbol{S}_{\boldsymbol{\theta}}\boldsymbol{p}_{\boldsymbol{\theta}}, \qquad \frac{\mathrm{d}s_{\boldsymbol{\theta}}^{\langle i,j\rangle}}{\mathrm{d}t} = \frac{\lambda^2(\xi)}{\kappa_{\boldsymbol{\theta}}^{\langle i,j\rangle}}\left[\frac{p_{\theta_i}p_{\theta_j}}{m_{\theta_i}} - T\delta_{ij}\right],$$

$$\frac{\mathrm{d}\xi}{\mathrm{d}t} = \frac{p_{\xi}}{m_{\xi}}, \qquad \frac{\mathrm{d}p_{\xi}}{\mathrm{d}t} = -\lambda'(\xi)\tilde{U}(\boldsymbol{\theta}) - W'(\xi) - [\lambda'(\xi)]^2 s_{\xi}p_{\xi}, \qquad \frac{\mathrm{d}s_{\xi}}{\mathrm{d}t} = \frac{[\lambda'(\xi)]^2}{\kappa_{\xi}}\left[\frac{p_{\xi}^2}{m_{\xi}} - T\right], \qquad (5)$$

where $\boldsymbol{S}_{\boldsymbol{\theta}}$ and $s_{\xi}$ denote the Nosé-Hoover thermostats coupled with $\boldsymbol{\theta}$ and $\xi$. Specifically, $\boldsymbol{S}_{\boldsymbol{\theta}} = \left[s_{\boldsymbol{\theta}}^{\langle i,j\rangle}\right]$ is a $D \times D$ matrix with the $(i,j)$-th elements $s_{\boldsymbol{\theta}}^{\langle i,j\rangle}$ dependent upon the multiplicative term $p_{\theta_i}p_{\theta_j}/m_{\theta_i}$. $\kappa_{\boldsymbol{\theta}}^{\langle i,j\rangle}$ and $\kappa_{\xi}$ are constants that denote the "thermal inertia" corresponding to $s_{\boldsymbol{\theta}}^{\langle i,j\rangle}$ and $s_{\xi}$, respectively. Intuitively, the thermostats $\boldsymbol{S}_{\boldsymbol{\theta}}$ and $s_{\xi}$ act as negative feedback controllers on the momenta $\boldsymbol{p}_{\boldsymbol{\theta}}$ and $p_{\xi}$. Consider the dynamics of $s_{\xi}$ in Eq. (5), when $p_{\xi}^2/m_{\xi}$ exceeds the reference $T$, the thermostat $s_{\xi}$ will increase, leading to a greater friction $-s_{\xi}p_{\xi}$ in updating $p_{\xi}$; the friction in turn reduces the magnitude of $p_{\xi}$, resulting in a decrease in the value of $p_{\xi}^2/m_{\xi}$. The negative feedback loop is thus established. With the help of thermostats, the noise injected into the system can be adaptively neutralised.

We define the diffusion coefficients $b_U(\boldsymbol{\theta}) := v_U(\boldsymbol{\theta})\,\mathrm{d}t/2$ and $\boldsymbol{B}_f(\boldsymbol{\theta}) = \left[b_f^{\langle i,j\rangle}(\boldsymbol{\theta})\right] := \boldsymbol{V}_f(\boldsymbol{\theta})\,\mathrm{d}t/2$ such that the variances $v_U(\boldsymbol{\theta})$ and $\boldsymbol{V}_f(\boldsymbol{\theta})$ of the mini-batch approximations evaluated at each of the discrete iterations can be embedded in the Fokker-Planck equation (FPE) [20] established in continuous time. FPE translates the microscopic motion of particles, formulated by SDEs, into the macroscopic time evolution of the state distribution in the form of PDEs. With FPE leveraged, we establish the theorem as follows to characterise the invariant distribution:

**Theorem 1.** *The system governed by the dynamics in Eq.* (5) *has the invariant distribution:*

$$\pi(\boldsymbol{\Gamma}, \boldsymbol{S}_{\boldsymbol{\theta}}, s_{\xi}) \propto e^{-\left[H(\boldsymbol{\Gamma}) + \left(s_{\xi} - \frac{b_U(\boldsymbol{\theta})}{m_{\xi}T}\right)^2 \kappa_{\xi}/2 + \sum_{i,j}\left(s_{\boldsymbol{\theta}}^{\langle i,j\rangle} - \frac{b_f^{\langle i,j\rangle}(\boldsymbol{\theta})}{m_{\theta_j}T}\right)^2 \kappa_{\boldsymbol{\theta}}^{\langle i,j\rangle}/2\right]/T}, \qquad (6)$$

*where* $\boldsymbol{\Gamma} = (\boldsymbol{\theta}, \xi, \boldsymbol{p}_{\boldsymbol{\theta}}, p_{\xi})$ *denotes the extended state as presented in Eq.* (4).

*Proof.* Recall FPE in its vector form [20]:

$$\frac{\partial}{\partial t}\pi(\boldsymbol{x},t) = -\frac{\partial}{\partial \boldsymbol{x}} \cdot \left[\boldsymbol{\mu}_x(\boldsymbol{x},t)\pi(\boldsymbol{x},t)\right] + \left[\frac{\partial}{\partial \boldsymbol{x}}\frac{\partial^\top}{\partial \boldsymbol{x}}\right] \cdot \left[\boldsymbol{B}_x(\boldsymbol{x},t)\pi(\boldsymbol{x},t)\right], \tag{7}$$

where $\boldsymbol{x} = \mathrm{vec}(\boldsymbol{\Gamma}, \boldsymbol{S}_\theta, s_\xi)$ denotes the vectorisation of the collection of all variables defined in Eq. (6), $\boldsymbol{\mu}_x$ and $\boldsymbol{B}_x$ represent the drift and diffusion terms associated with the dynamics in Eq. (5), respectively, and the dot operator $\cdot$ defines the composition of summation after element-wise multiplication.

We substitute the corresponding elements within Eq. (5) into the drift and diffusion of FPE in Eq. (7). As we presume that the introduced thermostats are mutually independent with each other, the invariant distribution can hence be factorised into marginals as $\pi(\boldsymbol{x}) = \pi_\Gamma \pi_{s_\xi} \prod_{i,j} \pi_{s_\theta^{\langle i,j\rangle}}$. It is straightforward to verify that those deterministic parts with the dependency only on $\boldsymbol{\Gamma}$ cancel exactly with each other. The remnants are the stochastic parts as well as the deterministic ones that depend on the thermostats $\boldsymbol{S}_\theta$ and $s_\xi$, which can be formulated as

$$\frac{\partial}{\partial t}\pi(\boldsymbol{x},t) = \frac{\partial}{\partial p_\xi}\left[\left[\lambda'(\xi)\right]^2 s_\xi p_\xi \pi\right] - \sum_{i,j}\frac{\partial}{\mathrm{d}s_\theta^{\langle i,j\rangle}}\left[\frac{\lambda^2(\xi)}{\kappa_\theta^{\langle i,j\rangle}}\left[\frac{p_{\theta_i}p_{\theta_j}}{m_{\theta_i}} - T\delta_{ij}\right]\pi\right] + \frac{\partial^2}{\partial p_\xi}\left[\left[\lambda'(\xi)\right]^2 b_U(\boldsymbol{\theta})\pi\right]$$

$$+ \frac{\partial}{\partial \boldsymbol{p}_\theta}\cdot\left[\lambda^2(\xi)\boldsymbol{S}_\theta \boldsymbol{p}_\theta \pi\right] - \frac{\partial}{\partial s_\xi}\left[\frac{\left[\lambda'(\xi)\right]^2}{\kappa_\xi}\left[\frac{p_\xi^2}{m_\xi} - T\right]\pi\right] + \left[\frac{\partial}{\partial \boldsymbol{p}_\theta}\frac{\partial^\top}{\partial \boldsymbol{p}_\theta}\right]\cdot\left[\lambda^2(\xi)\boldsymbol{B}_f(\boldsymbol{\theta})\pi\right]. \tag{8}$$

We solve for the invariant distribution $\pi(\boldsymbol{x})$ by equating Eq. (8) to zero. The resulted formulae for the marginals $\pi_{s_\xi}$ and $\pi_{s_\theta^{\langle i,j\rangle}}$ are obtained under the assumption of factorisation in the form of

$$\frac{1}{\pi_{s_\xi}}\frac{\partial \pi_{s_\xi}}{\partial s_\xi} = -\frac{\kappa_\xi}{T}\left[s_\xi - \frac{b_U(\boldsymbol{\theta})}{m_\xi T}\right] \quad \text{and} \quad \frac{1}{\pi_{s_\theta^{\langle i,j\rangle}}}\frac{\partial \pi_{s_\theta^{\langle i,j\rangle}}}{\partial s_\theta^{\langle i,j\rangle}} = -\frac{\kappa_\theta^{\langle i,j\rangle}}{T}\left[s_\theta^{\langle i,j\rangle} - \frac{b_f^{\langle i,j\rangle}(\boldsymbol{\theta})}{m_{\theta_j}T}\right]. \tag{9}$$

The solutions to Eq. (9) are clear: both $\pi_{s_\xi}$ and $\pi_{s_\theta^{\langle i,j\rangle}}$ are Gaussian distributions determined uniquely by the coefficients. The marginals $\pi_{s_\xi}$ and $\pi_{s_\theta^{\langle i,j\rangle}}$, along with the canonical distribution $\pi_\Gamma$ w.r.t. $H(\boldsymbol{\Gamma})$, constitute the invariant distribution defined in Eq. (6). □

Theorem 1 states that, when the system reaches equilibrium, the system state is distributed as Eq. (6), and the mini-batch noise is absorbed into the thermostats from the system dynamics in Eq. (5). Thus, we can marginalise out both $\boldsymbol{S}_\theta$ and $s_\xi$ to drop the noise, and then obtain the canonical distribution in Eq. (3). As we are seeking for the recovery of the target posterior from the canonical distribution, we can assign specific values to the tempering variable $\xi = \xi^*$ such that the effective temperature of the original system is held fixed at unity $T/\lambda(\xi^*) = 1$. Hence, the posterior $\rho(\boldsymbol{\theta}|\mathscr{D})$ equals to the marginal distribution w.r.t. $\boldsymbol{\theta}$ given $\xi^*$ satisfying $\lambda(\xi^*) = T$, which is obtained by the marginalisation of $\boldsymbol{p}_\theta$ and $p_\xi$ over the canonical distribution as follows:

$$\pi(\boldsymbol{\theta}|\xi^*) = \sum_{\boldsymbol{p}_\theta, p_\xi}\pi(\boldsymbol{\Gamma}|\xi^*) = \frac{\sum_{\boldsymbol{p}_\theta, p_\xi} e^{-H(\boldsymbol{\Gamma}|\xi^*)/T}}{\sum_{\boldsymbol{\Gamma}\backslash\xi} e^{-H(\boldsymbol{\Gamma}|\xi^*)/T}} = \frac{e^{-U(\boldsymbol{\theta})}}{\sum_{\boldsymbol{\theta}} e^{-U(\boldsymbol{\theta})}} = \frac{1}{Z_\theta(T)}e^{-U(\boldsymbol{\theta})} = \rho(\boldsymbol{\theta}|\mathscr{D}),$$

where $H(\boldsymbol{\Gamma}|\xi^*) = \lambda(\xi^*)U(\boldsymbol{\theta}) + W(\xi^*) + \boldsymbol{p}_\theta^\top \boldsymbol{M}_\theta^{-1}\boldsymbol{p}_\theta/2 + p_\xi^2/2m_\xi$ represents the extended Hamiltonian conditioning on the tempering variable $\xi = \xi^*$, when $\lambda(\xi^*) = T$ holds.

## 3.2 Tempering enhancement via adaptive biasing force

A necessary condition for the tempering scheme to be well-functioning is that the tempering variable $\xi$ can properly explore the majority of the domain of the coupling function $\mathrm{dom}\lambda(\xi)$; this ensures the expected variation on the effective temperature during sampling. For complex systems, however, it is often the case that the tempering variable is subject to a strong instantaneous force that prevents $\xi$ from proper exploration of $\mathrm{dom}\lambda(\xi)$ and therefore hinders the efficiency of tempering. The adaptive biasing force (ABF) algorithm [3] has emerged as a promising solution to such problem ever since its inception [4], where it was introduced to address the problem on fast calculation of the free energy of complex chemical or biochemical systems. Intuitively, ABF maintains and updates an estimate of the average force, i.e. the average of the instantaneous force exerted on the target variable. It then applies the estimate to the target variable in the opposite direction to counteract the instantaneous force and reduce it into small zero-mean fluctuations so that the variable undergoes random walks.

**Algorithm 1** Thermostat-assisted continuously-tempering Hamiltonian Monte Carlo

---

**Input:** stepsize $\eta_\theta, \eta_\xi$; level of injected noise $c_\theta, c_\xi$; thermal inertia $\gamma_\theta, \gamma_\xi$; # of steps for unit interval $K$

1: $r_\theta \sim \mathcal{N}(0, \eta_\theta I)$ and $r_\xi \sim \mathcal{N}(0, \eta_\xi)$; $(z_\theta, z_\xi) \leftarrow (c_\theta, c_\xi)$
2: INITIALISE( $\theta, \xi, $ abf, samples )
3: **for** $k = 1, 2, 3, \ldots$ **do**
4: $\quad \lambda \leftarrow$ LAMBDA( $\xi$ ); $\quad \delta\lambda \leftarrow$ LAMBDADERIVATIVE( $\xi$ )
5: $\quad z_\xi \leftarrow z_\xi + \delta\lambda^2 \left[ r_\xi^2 - \eta_\xi \right] / \gamma_\xi$
6: $\quad z_\theta \leftarrow z_\theta + \lambda^2 \left[ r_\theta^\top r_{\theta_j} / \dim(r_\theta) - \eta_\theta \right] / \gamma_\theta$
7: $\quad \mathcal{S} \leftarrow$ NEXTBATCH( $\mathcal{D}, k$ ); $\quad \delta A \leftarrow$ abf[ ABFINDEXING( $\xi$ ) ]
8: $\quad \tilde{U} \leftarrow$ MODELFORWARD( $\theta, \mathcal{S}$ ); $\quad \tilde{f} \leftarrow$ MODELBACKWARD( $\theta, \mathcal{S}$ )
9: $\quad r_\xi \leftarrow r_\xi - \delta\lambda \left[ \eta_\xi \tilde{U} + \mathcal{N}(0, 2c_\xi \eta_\xi) \right] - \delta\lambda^2 z_\xi r_\xi + \eta_\xi \delta A$
10: $\quad r_\theta \leftarrow r_\theta + \lambda \left[ \eta_\theta \tilde{f} + \mathcal{N}(0, 2c_\theta \eta_\theta I) \right] - \lambda^2 z_\theta r_\theta$
11: $\quad$ ABFUPDATE( abf$, \xi, \delta\lambda, \tilde{U}, k$ )
12: $\quad \xi \leftarrow \xi + r_\xi$
13: $\quad$ **if** ISINSIDEWELL( $\xi$ ) = false **then** $\qquad\qquad\qquad$ ▷ $\xi$ is restricted by the *well* of infinite height.
14: $\quad\quad r_\xi \leftarrow -r_\xi; \quad \xi \leftarrow \xi + r_\xi$ $\qquad\qquad\qquad\qquad$ ▷ $\xi$ bounces back when hitting the wall.
15: $\quad \theta \leftarrow \theta + r_\theta$
16: $\quad$ **if** $k = 0 \mod K$ and $\lambda = 0$ **then**
17: $\quad\quad$ APPEND( samples$, \theta$ ) $\qquad\qquad\qquad\qquad$ ▷ $\theta$ is collected as a new sample in samples.
18: $\quad\quad r_\theta \sim \mathcal{N}(0, \eta_\theta I)$ and $r_\xi \sim \mathcal{N}(0, \eta_\xi)$ $\qquad\qquad$ ▷ $r_\theta, r_\xi$ is *optionally* resampled.
19: **function** ABFUPDATE( abf$, \xi, \delta\lambda, \tilde{U}, k$ )
20: $\quad j \leftarrow$ ABFINDEXING( $\xi$ ) $\qquad\qquad\qquad\qquad$ ▷ $\xi$ is mapped to the index $j$ of the associated bin.
21: $\quad$ abf[ $j$ ] $\leftarrow [1 - 1/k]$abf[ $j$ ] $+ [1/k]\delta\lambda \cdot \tilde{U}$

---

Formally, the function of free energy w.r.t. $\xi$ is defined by convention in the form of

$$A(\xi) = -T \log \pi(\xi) + \text{const}, \quad \text{where } \pi(\xi) = \sum\nolimits_{\Gamma \setminus \xi} \pi(\Gamma) \text{ with the extended state } \Gamma = (\theta, \xi, p_\theta, p_\xi).$$

The equation of $p_\xi$ in Eq. (5) is then augmented with the derivative of $A(\xi)$ such that

$$\mathrm{d}p_\xi \big/ \mathrm{d}t = -\lambda'(\xi)\tilde{U}(\theta) - W'(\xi) + A'(\xi) - \left[ \lambda'(\xi) \right]^2 s_\xi p_\xi, \tag{10}$$

where $A'(\xi)$ is referred to as the adaptive biasing force induced by the free energy as

$$A'(\xi) = -\frac{T}{\pi(\xi)} \frac{\mathrm{d}\pi}{\mathrm{d}\xi} = \frac{\sum_{\Gamma \setminus \xi} \left[ \frac{\partial H}{\partial \xi} \right] e^{-H(\Gamma)/T}}{\sum_{\Gamma \setminus \xi} e^{-H(\Gamma)/T}} := \left\langle \frac{\partial H}{\partial \xi} \Big| \xi \right\rangle. \tag{11}$$

The brackets $\langle \cdot | \xi \rangle$ denote the conditional average, i.e. the average on the canonical distribution $\pi(\Gamma)$ with $\xi$ held fixed. $A'(\xi)$ is the average of the reversed instantaneous force on $\xi$. It is proved [14] that ABF converges to the equilibrium at which $\xi$'s free energy landscape is flattened, even though the augmentation in Eq. (10) alters the equations of motion originally defined in Eq. (5).

### 3.3 Implementation

As proved in Theorem 1, the dynamics in Eq. (5) is capable of preserving the correct distribution in the presence of noise. In principle, it requires the thermostat $S_\theta$ to be of size $D^2$ for the $D$-dimensional parameter $\theta$; however, the storage is unaffordable for complex models in high dimensions. A plausible option to mitigate this issue is to assume homogeneous $\theta$ and isotropic Gaussian noise such that the mass $M_\theta = m_\theta I$ and the variance $V_f(\theta) = v_f(\theta)I$; this simplifies the high-dimensional $S_\theta$ to scalar $s_\theta$. The confining potential $W(\xi)$ that determines the range of the tempering variable $\xi$ is implemented as a well of infinite height. When colliding with the boundary of $W(\xi)$, $\xi$ bounces back elastically with the velocity reversed. The Euler's method is then applied such that $\mathrm{d}t \to \Delta t$.

In Eq. (11), the calculation of $A'(\xi)$ involves the ensemble average $\langle \partial H / \partial \xi | \xi \rangle$, hence being intractable. Here we instead calculate the time average $\sum_k \partial H / \partial \xi |_{\xi_k}$, which is equivalent to the ensemble average in the long-time limit under the assumption of ergodicity; it can be readily calculated in a recurrent form during sampling. To maintain the runtime estimates of $A'(\xi)$, the range of $\xi$ is divided uniformly into $J$ bins of equal length with memory initialised in each of those bins. At each time step $k$, ABF determines the index $j$ of the bin in which the tempering variable $\xi = \xi_k$ is currently located, and then updates the time average using the record in memory and the current force $\partial H / \partial \xi |_{\xi_k}$ evaluated at $\xi_k$.

With all components assembled, we establish the TACT-HMC algorithm as Algorithm. 1 with

$$r_\theta = \frac{p_\theta \Delta t}{m_\theta}, \quad r_\xi = \frac{p_\xi \Delta t}{m_\xi}, \quad z_\theta = s_\theta \Delta t, \quad z_\xi = s_\xi \Delta t, \quad \eta_\theta = \frac{\Delta t^2}{m_\theta}, \quad \eta_\xi = \frac{\Delta t^2}{m_\xi}, \quad \gamma_\theta = \frac{\kappa_\theta}{m_\theta D}, \quad \gamma_\xi = \frac{\kappa_\xi}{m_\xi}$$

applied as the change of variables for the convenience of implementation. Furthermore, we introduce additional Gaussian noises $\mathcal{N}(0, 2c_\xi \eta_\xi)$ and $\mathcal{N}(0, 2c_\theta \eta_\theta I)$ in momenta updates to improve ergodicity.

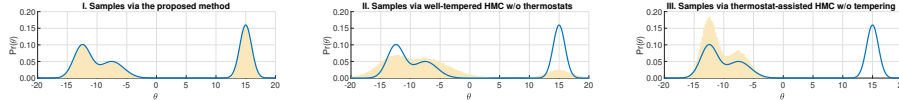

(a) Histograms of samples generated by TACT-HMC and the ablated alternatives, with the target shown in blue.

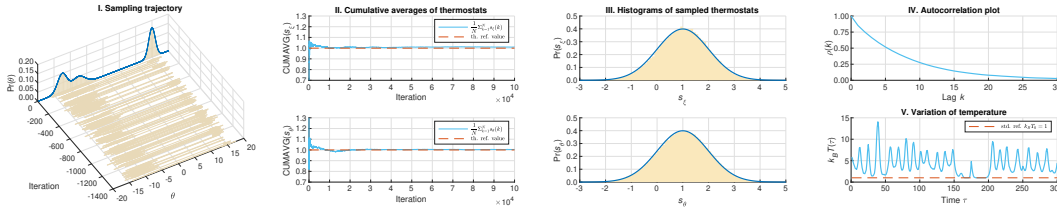

(b) I: Sampling trajectory of TACT-HMC, demonstrating robust mixing property; II: Cumulative averages of thermostats, indicating fast convergence to the theoretical reference values drawn by red lines; III: Histograms of sampled thermostats, showing a good fit to the theoretical distributions by blue curves; IV: Autocorrelation plot of samples, the decreasing of autocorrelation is comparably fast; V: (A snapshot of) variation on the effective system temperature during simulation, with the standard reference of unity temperature marked by red line.

Figure 1: Experiment on sampling a $1d$ synthetic distribution.

## 4    Related work

Since the inception of the stochastic gradient Langevin dynamics (SGLD) [24], algorithms originated from stochastic approximation [21] have received increasing attention on tasks of Bayesian learning. By adding the right amount of noise to the updates of the stochastic gradient descent (SGD), SGLD manages to properly sample the posterior in a random-walk fashion akin to the full-batch Metropolis-adjusted Langevin algorithm (MALA) [22]. To enable the Hamiltonian dynamics for efficient state space exploration, Chen et al. [2] extended the mechanism designed for SGLD to HMC, and proposed the stochastic gradient Hamiltonian Monte Carlo (SGHMC). As is shown that the stochastic gradient drives the Hamiltonian dynamics to deviate, SGHMC estimates the unknown noise from the stochastic gradient with the Fisher information matrix, and then compensates the estimated noise by augmenting the Hamiltonian dynamics with an additive friction derived from the estimated Fisher matrix. It turns out that the friction can be linked to the momentum term within a class of accelerated gradient-based methods [19, 17, 23] in optimisation. Shortly after SGHMC, Ding et al. [5] came up with the idea of incorporating the Nosé-Hoover thermostat [18, 11] into the Hamiltonian dynamics in replacement of the constant friction in SGHMC, and hence developed the stochastic gradient Nosé-Hoover thermostat (SGNHT). The thermostat in SGNHT serves as an adaptive friction which adaptively neutralises the mini-batch noise from the stochastic gradient into the system [12].

Parallel to those aforementioned studies, recent advances in the development of continuous tempering [8, 15] as well as its applications in machine learning [25, 9] are of particular interest. Ye et al. [25] proposed the continuously tempered Langevin dynamics (CTLD), which leverages the mechanism of continuous tempering and embeds the tempering variable in an extended stochastic gradient second-order Langevin dynamics. CTLD facilitates exploration on rugged landscapes of objective functions, locating the "good" wide valleys on the landscape and preventing early trapping in the "bad" narrow local minima. Nevertheless, CTLD is designed to be an initialiser for training deep neural networks; it serves as an enhancement of the subsequent gradient-based optimisers instead of a Bayesian solution. From the Bayesian perspective, Graham et al. [9] developed the continuously-tempered Hamiltonian Monte Carlo (CTHMC) operating in a full-batch setting. CTHMC augments the Hamiltonian system with an extra continuously-varying control variate borrowed from the scheme of continuous tempering, which enables the extended Hamiltonian dynamics to bridge between sampling a complex multimodal target posterior and a simpler unimodal base distribution. Albeit beneficial for mixing, its dynamics lacks the ability to handle the mini-batch noise, and thus fails to function properly with mini-batches.

## 5    Experiment

Two sets of experiments are carried out. We first conduct an ablation study with synthetic distributions, where we visualise the system dynamics and validate the efficacy of TACT-HMC. We then evaluate the performance of our method against several strong baselines on three real datasets.

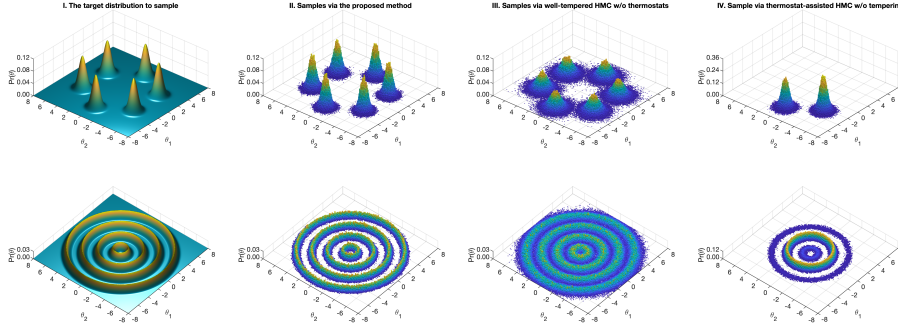

Figure 2: Experiments on sampling two $2d$ synthetic distributions. *Left*: The distributions to sample; *Mid-left*: Histograms sampled by TACT-HMC; *Mid-right*: Histograms by the well-tempered sampler without thermostatting; *Right*: Histograms by the thermostat-assisted sampler without tempering.

## 5.1 Multimodal sampling of synthetic distributions

We run TACT-HMC on three $1d/2d$ synthetic distributions. In the meantime, two ablated alternatives are initiated in parallel with the same setting: one is equipped with thermostats but without tempering for sampling acceleration, the other is well-tempered but without thermostatting against noise. The distributions are synthesised to contain multiple distant modes; the calculation of gradient is perturbed by Gaussian noise that is unknown to all samplers.

Figure 1 summarises the result of sampling a mixture of three $1d$ Gaussians. As the figure indicates, only TACT-HMC is capable of correctly sampling from the target. The sampler without thermostatting is heavily influenced by the noise in gradient, resulting in a spread histogram; while the one without tempering gets trapped by those energy barriers and hence fails to explore the entire space of system configurations. The sampling trajectory and properties of TACT-HMC are illustrated in details in Fig. 1b, which justifies the correctness of TACT-HMC. The autocorrelation of samples $\rho(k)$ is calculated and shown in Fig. 1b(IV), which decreases monotonically from $\rho(0) = 1$ down to $\rho(\infty) \to 0^+$. The effective sample size (ESS) can thus be readily evaluated through the formula

$$ESS = \frac{n}{1 + 2 \sum_{k=1}^{\infty} \rho(k)}, \quad \text{with } \rho(k) \text{ as the autocorrelation at lag } k.$$

The ESS for TACT-HMC in this $1d$ Gaussian mixture case is $2.1096 \times 10^4$ out of $n = 10^5$ samples, which is roughly 60.2% of the value for SGHMC and 50.9% of that for SGNHT. We believe that the non-linear interaction between the parameter of interest $\theta$ and the tempering variable $\xi$ via the multiplicative term $\lambda(\xi)U(\theta)$ results in a longer autocorrelation time and hence a lower ESS value. We also investigate the variation of the effective system temperature during sampling. A snapshot of the trajectory regarding the effective system temperature is demonstrated in Fig. 1b(V): it constantly oscillates between higher and lower temperatures, and returns to the unity temperature occasionally.

We further conduct two $2d$ sampling experiments as shown in Fig. 2. Comparing between columns, we find that TACT-HMC recovers those multiple modes for both distributions while neutralising the influence of the noise in gradient; however, the samplings by the ablated alternatives are impaired either by the noise in gradient or by the energy barriers as discovered in the $1d$ scenario.

## 5.2 Bayesian learning on real datasets

Stepping out of the study on the synthetic cases, we then move on to the tasks of image classification on three real datasets: EMNIST[3], Fashion-MNIST[4] and CIFAR-10. The performance is evaluated and compared in terms of the accuracy of classification on three types of neural networks: multilayer perceptrons (MLPs), recurrent neural networks (RNNs), and convolutional neural networks (CNNs). Two recent samplers are chosen as part of the baselines, namely SGNHT [5] and SGHMC [2]; besides, two widely-used gradient-based optimisers, Adam [13] and momentum SGD [23], are compared.

Each method will keep running for 1000 epochs in either sampling or training before the evaluation and comparison. We further apply random permutation to a certain percentage (0%, 20%, and 30%) of the training labels at the beginning of each epoch for demonstrating the robustness of our method.

Table 1: Result of Bayesian learning experiments on real datasets

| % permuted labels | MLP on EMNIST | | | RNN on Fashion-MNIST | | | CNN on CIFAR-10 | | |
|---|---|---|---|---|---|---|---|---|---|
| | 0% | 20% | 30% | 0% | 20% | 30% | 0% | 20% | 30% |
| Adam [13] | 83.39% | 80.27% | 80.63% | 88.84% | 88.35% | 88.25% | 69.53% | 72.39% | 71.05% |
| momentum SGD [23] | 83.95% | 82.64% | 81.70% | 88.66% | 88.91% | 88.34% | 64.25% | 65.09% | 67.70% |
| SGHMC [2] | 84.53% | 82.62% | 81.56% | 90.25% | 88.98% | 88.49% | 76.44% | 73.87% | 71.79% |
| SGNHT [5] | 84.48% | 82.63% | 81.60% | 90.18% | 89.10% | 88.58% | 76.60% | 73.86% | 71.37% |
| TACT-HMC (Alg. 1) | **84.85**% | **82.95**% | **81.77**% | **90.84**% | **89.61**% | **89.01**% | **78.93**% | **74.88**% | **73.22**% |

All four baselines are tuned to their best on each task; the setting of TACT-HMC will be specified for each task in the corresponding subsection. For the baseline samplers, the accuracy of classification is calculated from Monte Carlo integration on all sampled models; for the baseline optimisers, the performance is evaluated directly on test sets after training. The result is summarised in Table. 1.

**EMNIST classification with MLP.** The MLP herein defines a three-layered fully-connected neural network with the hidden layer consisting of 100 neurons. EMNIST Balanced is selected as the dataset, where 47 categories of images are split into a training set of size 112,800 and a complementary test set of size 18,800. The batch size is fixed at 128 for all methods in both sampling and training tests. For readability, we introduce a 7-tuple $[\eta_\theta, \eta_\xi, c_\theta, c_\xi, \gamma_\theta, \gamma_\xi, K]$ as the specification to set up TACT-HMC (see Alg. 1). In this specification, $[\eta_\theta, c_\theta, \gamma_\theta]$ denote the step size, the level of the injected Gaussian noise and the thermal inertia, all w.r.t. the parameter of interest $\theta$; similarly, $[\eta_\xi, c_\xi, \gamma_\xi]$ represent the quantities corresponding to the tempering variable $\xi$; $K$ defines the number of steps in simulating a unit interval. In this experiment, TACT-HMC is configured as $[0.0015, 0.0015, 0.05, 0.05, 1.0, 1.0, 50]$.

**Fashion-MNIST classification with RNN.** The RNN contains a LSTM layer [10] as the first layer, with the input/output dimensions being $28/128$. It takes as the input via scanning a $28 \times 28$ image vertically each line of a time. After 28 steps of scanning, the LSTM outputs a representative vector of length 128 into ReLU activation, which is followed by a dense layer of size 64 with ReLU activation. The prediction regarding 10 categories is generated through softmax activation in the output layer. The batch size is fixed at 64 for all methods in comparison. The specification of TACT-HMC in this experiment is determined as $[0.0012, 0.0012, 0.15, 0.15, 1.0, 1.0, 50]$.

**CIFAR-10 classification with CNN.** The CNN comprises of four learnable layers: from the bottom to the top, a $2d$ convolutional layer using the kernel of size $3 \times 3 \times 3 \times 16$, and another $2d$ convolutional layer with the kernel of size $3 \times 3 \times 16 \times 16$, then two dense layers of size 100 and 10. ReLU activations are inserted between each of those learnable layers. For each convolutional layer, the stride is set to $1 \times 1$, and a pooling layer with $2 \times 2$ stride is appended after the ReLU activation. Softmax function is applied for generating the final prediction over 10 categories. The batch size is fixed at 64 for all methods. Here, TACT-HMC's specification is set as $[0.0010, 0.0010, 0.10, 0.10, 1.0, 1.0, 50]$.

**Discussion.** As summarised in Table. 1, TACT-HMC outperforms all four baselines on the accuracy of classification. Specifically, TACT-HMC demonstrates advantages on complicated tasks, e.g. the CIFAR-10 classification with CNN where the model has relatively higher complexity and the dataset contains multiple channels. For the RNN task, our method outperforms others with roughly 0.5% on accuracy. The performance gain on the MLP task is rather limited; we believe the reason for this is that the complexities of both model and dataset are essentially moderate. When the random permutation is applied to a larger portion of training labels, TACT-HMC still maintains robust performance on the accuracy of classification, even though the landscape of the objective function becomes rougher and the system dynamics gathers more noise.

# 6 Conclusion

We developed a new sampling method, which is called the thermostat-assisted continuously-tempered Hamiltonian Monte Carlo, to facilitate Bayesian learning with large datasets and multimodal posterior distributions. The method builds a well-tempered Hamiltonian system by incorporating the scheme of continuous tempering in the system for classic HMC, and then simulates the dynamics augmented by Nosé-Hoover thermostats. This sampler is designed for two substantial demands: first, to efficiently generate representative i.i.d. samples from complex multimodal distributions; second, to adaptively neutralise the noise arising from mini-batches. Extensive experiments have been carried out on both synthetic distributions and real-world applications. The result validated the efficacy of tempering and thermostatting, demonstrating great potentials of our sampler in accelerating deep Bayesian learning.

## Footnotes

[3]https://www.nist.gov/itl/iad/image-group/emnist-dataset

[4]https://github.com/zalandoresearch/fashion-mnist

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
