[Supplementary Material]

# Thermostat-assisted continuously-tempered Hamiltonian Monte Carlo for Bayesian learning
## *Supplementary Information*

## A    Characteristics of coupling function $\lambda(\xi)$ and confining potential $W(\xi)$

This section serves as a supplementary specification for both $\lambda(\xi)$ and $W(\xi)$.

Let us first recall the Hamiltonian of the extended system; for readability, we rewrite Eq. (4) here:

$$H(\mathbf{\Gamma}) = \lambda(\xi)U(\boldsymbol{\theta}) + W(\xi) + \boldsymbol{p}_{\boldsymbol{\theta}}^{\top}\boldsymbol{M}_{\boldsymbol{\theta}}^{-1}\boldsymbol{p}_{\boldsymbol{\theta}}/2 + p_{\xi}^2/2m_{\xi},$$

where $\lambda(\xi) \in \mathbb{R}^+$ denotes the coupling function that maps the tempering variable $\xi \in \mathbb{R}$ to a multiplier to the (inverse) temperature so that the effective temperature $T/\lambda(\xi)$ for the original system can vary. $W(\xi)$ represents the confining potential for $\xi$, acting as a potential well to physically restrict $\xi$'s range. Note that the effective temperature of the original system $T/\lambda(\xi)$ depends on the temperature of the extended system $T$, which is constant. In our experiments, we fix it at $T = 1$.

An illustration of both the coupling function $\lambda(\xi)$ and the confining potential $W(\xi)$ is shown in Fig. 1. The potential $W(\xi)$, as explained in §3.3, is implemented as a well of infinite height, depicted as the vertical red dashed lines. The coupling function $\lambda(\xi)$, on the other hand, is a first-order differentiable function with a plateau at the level of $\lambda(\xi^*) = 1$, drawn as the blue curve. We highlight the plateau by the shaded region in Fig. 1. For simplicity, $\lambda(\xi)$ is built to be even, i.e. symmetric w.r.t. the $y$-axis.

Figure 1: Illustration of the coupling function $\lambda(\xi)$ and confining potential $W(\xi)$. The shaded region represents the interval of standard temperature, in which the sampler generates unbiased samples.

Specifically, we define the coupling function in the form of

$$\frac{1}{\lambda(\xi)} = \begin{cases} 1 & \text{if } |\xi| \leq \xi_0 \,, \\ 1 + \left(\frac{|\xi| - \xi_0}{\xi_1 - \xi_0}\right)^n & \text{if } |\xi| > \xi_0 \,, \end{cases} \quad \text{with } \xi_1 > \xi_0 > 0 \text{ and } n \in \mathbb{N}^+.$$

By definition, a plateau of width $2\xi_0$ is placed at the centre; $\lambda(\xi)$ decays monotonically as $|\xi|$ increases when $|\xi|$ exceeds $\xi_0$, and it approaches $0^+$ in the limit $|\xi| \to \infty$.

In the synthetic cases, we use the parameter $[\xi_0 = 1/3, \xi_1 = 1, n = 3]$ so that $\lambda(\xi) \equiv 1$ if $|\xi| \leq \xi_0 = 1/3$. It reaches the effective temperature of 2 when $|\xi| = \xi_1 = 1$; by placing the two walls of the potential well $W(\xi)$ at $|W_0| = \pm 5/3$, the highest temperature that can be reached is 9.

Given a well-tuned sampler, the tempering variable $\xi$ should be able to move freely within the well $W(\xi)$, i.e. the probability of finding $\xi$ in any interval with fixed length must be equal. In this scenario, the efficiency of obtaining an unbiased system configuration is equivalent to the probability of finding the system at standard temperature, which then equals to the chance of $\xi$ being found on the plateau:

$$\text{efficiency} = \frac{|\xi_0|}{|W_0|},$$

where the efficiency of sampling, in our setting, is 20%.

Here we would like to emphasise that the magnitude of $\lambda'(\xi)$ may have some influence in stability of simulating the dynamics in Eq. (5). The magnitude $|\lambda'(\xi)|$ is, however, in a way inversely proportional to the ratio $|\xi_0|/|W_0|$ given a specific form of $\lambda(\xi)$ with the highest temperature available to hold fixed (i.e. by keeping the functional of $\lambda(\xi)$ unchanged and fixing the value at $\lambda(\xi = W_0)$). For that reason, we suggest the efficiency not to exceed 25%.

## B    Alternative approach to tempering enhancement by *Metadynamics*

We have leveraged the adaptive biasing force (ABF) method [2] in Algorithm 1 in order to cancel the instantaneous force preventing the tempering variable $\xi$ from free motion. Equivalently, ABF flattens the actual potential that $\xi$ feels, which is essentially the superposition of the confining potential $W(\xi)$ and that arises from the interaction between $\xi$ and $\boldsymbol{\theta}$ through $\lambda(\xi)U(\boldsymbol{\theta})$.

Metadynamics [3] has emerged as an alternative to ABF for a similar purpose of enhancing sampling. It introduces a history-dependent repulsive biasing potential to the target variable, i.e. $\xi$, to discourage $\xi$ from revisiting the places it has already visited. Due to its simplicity and the robustness, this method has been widely used in a variety of disciplines, ranging from science to engineering.

To implement Metadynamics, we establish a history-dependent biasing potential $A(\xi, t)$ on $\xi$'s feasible interval confined by $W(\xi)$. The interval is then divided into $J$ bins with length $\delta$; for each bin $j$, we maintain and update a memory $A_j(t)$ stored at the centre $a_j$ of bin $j$ in the form

$$A_j(t) = \sum_{\tau=0}^{t} h_A \mathcal{I}\left[\xi(\tau) \in \text{bin } j\right],$$

where $\mathcal{I}[\xi(t) \in \text{bin } j] = 1$ if $\xi(t) \in \text{bin } j$ otherwise 0 represents the indicator function and $h_A$ defines the incremental for $A_j(t)$ in each of the updates. By tracking $\xi(t)$ in the runtime, we locate the current bin $j$ and then increase the previous value $A_j(t-1)$ in that bin $j$ by $h_A$.

The resulting biasing force can be readily calculated by

$$\partial_\xi A(\xi, t) = \begin{cases} \left[A_j(t) - A_{j+1}(t)\right]/\delta & \text{if } \xi \in \text{bin } j \text{ and } \xi \geq a_j, \\ \left[A_{j-1}(t) - A_j(t)\right]/\delta & \text{if } \xi \in \text{bin } j \text{ and } \xi < a_j. \end{cases}$$

A major problem that the vanilla Metadynamics [3] encounters is that it lacks a proof of convergence because the incremental $h_A$ remains constant; the biasing potential $A_j(t)$ grows proportionally to the time elapsed in simulation. Recent advances [1] in the development of Metadynamics seem to have mitigated this issue, which enables its application in a wider range of tasks [4].