[Reviews · NeurIPS 2018]

Reviewer 1



This paper explores the use of continuous tempering for sampling of multimodal posteriors using HMC. As pointed out by the authors, the main contribution lies in the experimental results, which show good results on real datasets. The exposition is clear most of the time, but one point is not clear to me. The target posterior is obtained for those samples where \xi=\xi*, such that T=\lamda(\xi*). But since \xi is continuous, in order for the set of points where \xi=\xi* holds to have non-zero probability, the function \lambda() should be flat and equal to T in some region. But no discussion is given of this point.

Reviewer 2



The paper proposes a novel sampling method using thermostat assisted continuous tempering for HMC. The proposed method helps to sample iid data from multi-modal distributions and neutralises biases from mini-batch samples. The paper is very well written and easy to follow. The experiment results are both visually appealing and positive on both the toy and real world datasets. It would be nice if the authors also measured the ESS of the samples and compared it to standard HMC, and preconditioned HMC.

Reviewer 3



This paper presents a sampling method that combines Hamiltonian Monte Carlo (HMC), mini-batches, tempering, and thermostats, to more efficiently explore multimodal target distributions. It is demonstrated on a number of substantial neural network problems using real data sets. This is an interesting method, and the empirical results are quite substantial. Figure 2 does a nice job of demonstrating how the omission of any of the ingredients (e.g. the tempering, or the thermostat) is detrimental to the overall result, which is a nice illustration of how the combination works together well. This is followed by some substantial image classification examples. I have one major, and one minor issue with the work, which perhaps the authors can address in their feedback: 1. It is not clear to me that the work contributes any new ideas beyond those already introduced in the following paper (cited as [5]): Ding et. al. (2014). Bayesian Sampling Using Stochastic Gradient Thermostats. Unless I am missing something, the above work introduces a sampler that combines the same components as listed above, and it, too, offers substantial empirical results. The authors might like to clarify what they see as the contribution of their work. 2. Given that these are Bayesian methods, I would have liked to have seen comparison of the posterior distributions or empirical convergence metrics, and not merely predictive performance, as given in Table 1. Presumably there is an "honest" predictive performance that is obtained by the exact posterior distribution for the model and data under study, and it could be obtained (hypothetically, at least) with any of the sampling methods (not the optimizations methods obviously) with a sufficient number of iterations. If a method has not converged sufficiently close to the posterior distribution, the predictive performance may be higher or lower than the honest value. It would not necessarily be lower. So these results are not really definitive as to which method is working best at obtaining the posterior distribution. To show that, it would be useful to see some conventional convergence diagnostics such as autocorrelation plots, effective sample size, etc. Or, if the argument is that the Bayesian approach offers a regularization against overfitting, rather than us caring about having a good estimate of the posterior distribution, some uncertainty range could be reported in Table 1 (posterior variance of accuracy, or 95% credibility over posterior samples, etc), to calibrate the significance of the increase in predictive performance for the method that is presented. (I presume the predictive performance reported is the mean accuracy over the posterior samples, given the comment at the bottom of page 7, "For the sampler baselines, the final accuracy is calculated from the Monte Carlo integration on all sampled models...") AFTER AUTHOR FEEDBACK I thank the authors for their detailed response. The authors have addressed my concerns regarding novelty. Having had a better look at Ding et al. now I am satisfied that the addition of continuous tempering is indeed novel, and have increased my score accordingly. The authors have suggested that they will look at some additional metrics, which I would also very much like to see.